# Fine-Scale Dasymetric Population Mapping with Mobile Phone and Building Use Data Based on Grid Voronoi Method

**Zhenghong Peng** [1], **Ru Wang** [1], **Lingbo Liu** [2,*] and **Hao Wu** [1]

[1] Department of Graphics and Digital Technology, School of Urban Design, Wuhan University, Wuhan 430072, China; pengzhenghong@whu.edu.cn (Z.P.); wang_ru@whu.edu.cn (R.W.); wh79@whu.edu.cn (H.W.)

[2] Department of Urban Planning, School of Urban Design, Wuhan University, Wuhan 430072, China

\* Correspondence: lingbo.liu@whu.edu.cn; Tel.: +86-27-6877-3062

**Abstract:** Fine-scale population mapping is of great significance for capturing the spatial and temporal distribution of the urban population. Compared with traditional census data, population data obtained from mobile phone data has high availability and high real-time performance. However, the spatial distribution of base stations is uneven, and the service boundaries remain uncertain, which brings significant challenges to the accuracy of dasymetric population mapping. This paper proposes a Grid Voronoi method to provide reliable spatial boundaries for base stations and to build a subsequent regression based on mobile phone and building use data. The results show that the Grid Voronoi method gives high fitness in building use regression, and further comparison between the traditional ordinary least squares (OLS) regression model and geographically weighted regression (GWR) model indicates that the building use data can well reflect the heterogeneity of urban geographic space. This method provides a relatively convenient and reliable idea for capturing high-precision population distribution, based on mobile phone and building use data.

**Keywords:** mobile phone data; building use data; dasymetric mapping; fine-scale population

## 1. Introduction

High-precision mapping of urban population plays an essential role in urban research, which is supportive for urban planning management [1], optimization of resource allocation [2], and understanding of urban spatial structure [3]. The dasymetric mapping method [4–10] has become the mainstream direction for high-precision population mapping, utilizing the correlation between population data and auxiliary data to disaggregate population data into micro-spatial levels through interpolation algorithm. With the rapid development of information technology in recent years, land use data [11,12], remote sensing data [13–18], and point of interest (POI) data [19] have become auxiliary data for dasymetric mapping, which has greatly improved the accuracy of population distribution.

However, the accuracy of dasymetric mapping is still susceptible to the resolution of source data and the correlation between auxiliary data and micro-scale population distribution. Most traditional studies take census data as the source population, which has the disadvantage of insufficient timeliness and cannot meet the current planning management requirements for dynamic high-precision population data [20]. Moreover, attributes of auxiliary data in different regions may be the same or similar, which cannot reflect spatial heterogeneity, and brings challenges to the improvement of method accuracy.

Recently, a considerable amount of literature has proposed the application of mobile phone data to study dynamic population distribution [21–25], commuting patterns [26,27], urban green space accessibility [28], and the urban spatial structure [29–31]. Mobile phone data characterized

by base stations can be more accurate and dynamic, reflecting the overall population distribution pattern [21,28,32]. Many articles have used mobile phone data as aggregate source data for population distribution calculation. However, there are still three main challenges with mobile phone data: (1) the number of mobile phone users is restricted by the market size of the operator and cannot reflect the real population; (2) the spatial distribution of mobile phone base stations is extensive and extremely uneven; and (3) the boundary range that the base station serves people remains high in uncertainty [28].

Furthermore, building data is suggested as the more suitable ancillary data for dasymetric population mapping, since buildings are the essential carrier of people's daily living [19,33–35]. Though most often use footprint and level data of buildings [36], the dynamic population is largely affected by buildings with different kinds of land use types. Thus, buildings with various functions may directly affect the population size of the surrounding base station.

This paper proposed a combined method of Grid Voronoi and building use regression (BUR) for fine-scale population mapping based on mobile phone data and land use integrated building data. This paper also compared traditional ordinary least squares (OLS) regression and geographically weighted regression (GWR) analysis to explore the correlation between buildings and population distribution. The results show that both of these show high correlations, and GWR shows little improvement, indicating that the building data can reflect the spatial heterogeneity. The mobile phone population was rescaled based on the census data, to eliminate the impact of the market share of operators, and the final grid population mapping of 1 km × 1 km was generated with the fine resolution level of buildings, which provides valuable spatiotemporal data for the study of urban structure characteristics.

## 2. Materials and Methods

### 2.1. Study Area

Wuhan is a city in Central China (Figure 1a), with a total land area of 8569.15 square kilometers. It is located in the east of Jianghan Plain and the middle reaches of the Yangtze River. The Yangtze River, the world's third-largest river, and its largest tributary, the Han River, run across the city center, dividing the central urban area of Wuhan into three parts, forming a pattern of three towns, Wuchang, Hankou, and Hanyang, which stand at the top of each other across the river. The study area is the Urban Development Zone (UDZ) of Wuhan. It is the main agglomeration area of urban functions, and the controlled expansion area of urban space (Figure 1b).

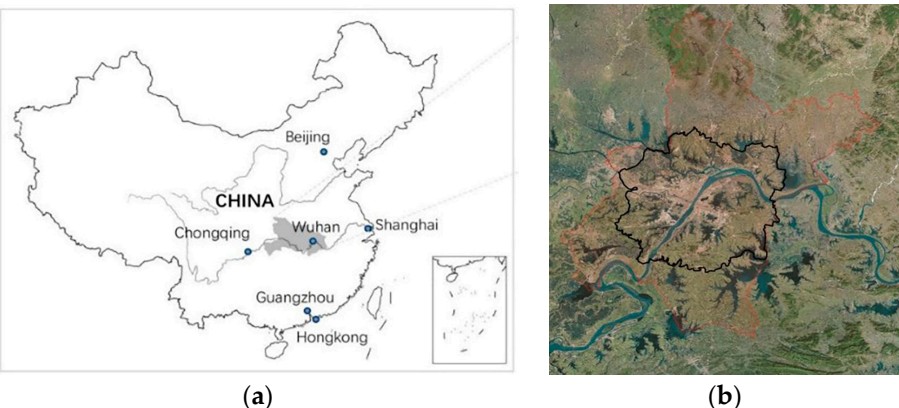

(a)　　　　　　　　　　　　　　　　　(b)

**Figure 1.** Map of the study area in Wuhan, China: (**a**) the geographic location of Wuhan, China; (**b**) Urban Development Zone (UDZ) of Wuhan.

## 2.2. Data and Preprocessing

The study used the call detail record (CDR) data of March 2016 in Wuhan city (Table 1). Steps are as follows to obtain the mobile phone population in the corresponding period. We used the non-work time population (defined as the resident population of a certain place) to build the building use regression (BUR) model.

- We matched the mobile phone number and the user ID to eliminate all private information, and then removed the invalid and noise data.
- We counted the base stations with the highest call frequency of users in the work time, non-work time, and all-time (Table 2), matching the base station code and the user ID, and summarized the number of users that the base station served at different periods.

**Table 1.** The call detail record data.

| User ID | The Starting Time of the User's Call | Base Station ID |
|---|---|---|
| 2700000127588 | 2016-03-09-9.15.24.000000 | 5760859194 |
| 2700000131734 | 2016-03-09-9.15.18.000000 | 2872636812 |
| 2700000122631 | 2016-03-09-9.15.49.000000 | 2893525929 |

**Table 2.** Period details.

| Time | Period |
|---|---|
| work time | Monday to Friday from 7:00 am to 7:00 pm |
| non-work time | Monday to Friday from 7:00 pm to 7:00 am, and Saturday and Sunday |
| all-time | Monday to Sunday |

Based on the 2016 land use data of the UDZ of Wuhan, provided by Wuhan Planning Institute (Figure 2a), we extracted nine types of land information: A (administrative and public service land), B (commercial service facility land), R (residential land), MW (industrial and logistics storage land), U (public facilities land), G (green land and square land), H (development and construction land), E (non-construction land, including waters, agricultural and forest land, open mining land, and abandoned land), and F (mixed land).

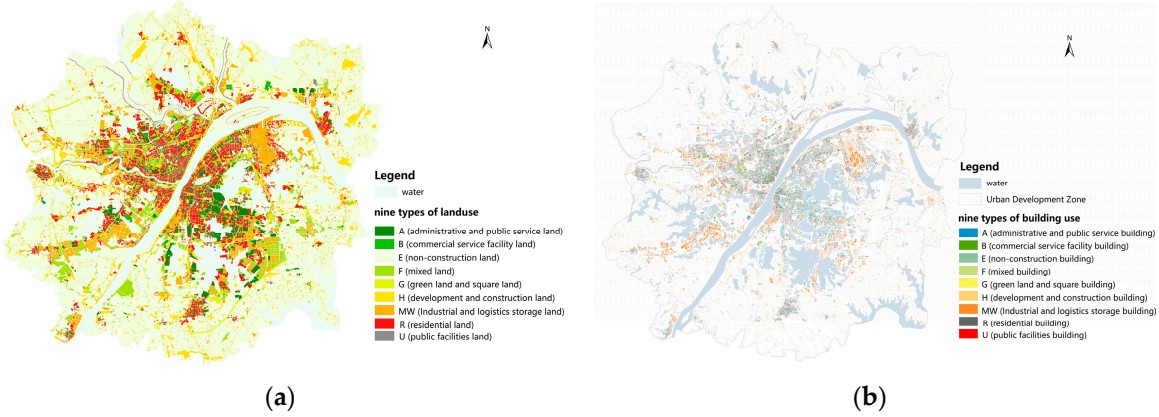

(**a**)                                    (**b**)

**Figure 2.** Data of the Urban Development Zone (UDZ) of Wuhan: (**a**) land use data in 2016; (**b**) building data in 2016.

Based on the 2016 building vector data of the UDZ of Wuhan, provided by Wuhan Planning Institute (Figure 2b), we edited the attribute table of building vector data and retained the attribute information, such as building floor and area. According to the spatial scope of buildings, we assigned

land properties to buildings to get the nine types of building use data: A (administration and public service building), B (commercial facilities building), R (residential building), MW (Industrial and logistics storage building), U (public facilities building), G (landscape building, such as green square), H (development building), E (a small number of agricultural buildings), and F (mixed function buildings).

### 2.3. Research Methods

#### 2.3.1. The Grid Voronoi Method

The study created 1 km × 1 km grids in the Urban Development Zone of Wuhan city to reduce the impact of the uneven spatial distribution of the original base stations, by aggregating the original base station points (Figure 3c). The total number of base stations was 30,725 (Figure 3a). The minimum value, maximum value, and average value in each grid were 0, 221, and 9, respectively (Figure 3b). We summarized the total number of non-work time populations within each grid and removed grids with zero population. The Grid Voronoi was generated by centroids of the retained grids, to define the boundary of the base station service range (Figure 3d). The mobile phone population and the total area of various buildings in each Voronoi were further summarized.

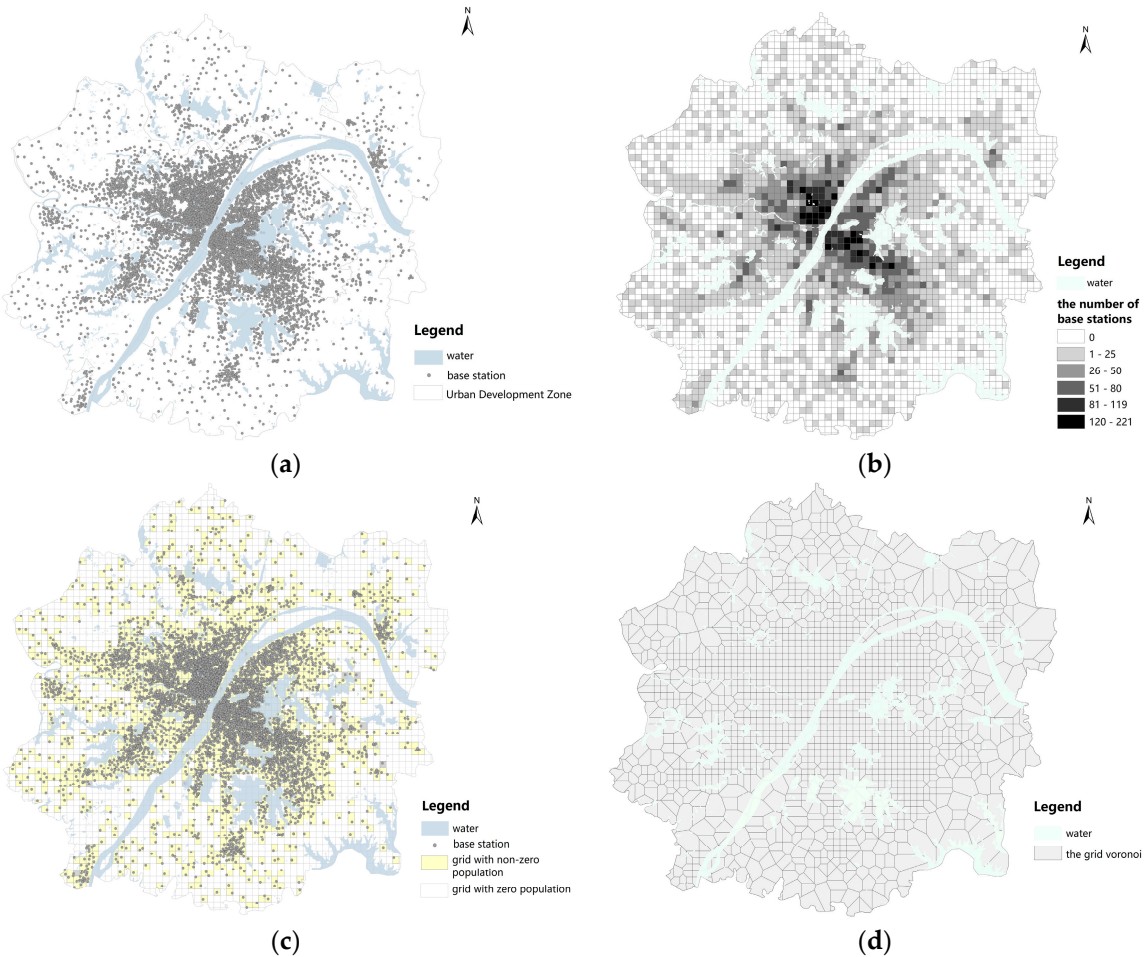

**Figure 3.** Process mapping of obtaining Grid Voronoi service area: (**a**) base station data; (**b**) number analysis of the gridded base station; (**c**) gridded base station data; (**d**) the Grid Voronoi.

### 2.3.2. Building Use Regression (BUR)

The research proposed a BUR model based on the land use regression (LUR) model. The LUR model was put forward by Briggs in 1997 [37] and has become one of the main application models of air pollution simulation research, by exploring the statistical linear correlation between the pollutant concentration in a specific range around the pollution detection station and its environmental conditions.

The BUR proposed in this study was to establish the OLS regression model within the service scope of the mobile base station to explore the correlation between population and building properties, and to observe the ability of building data to reflect spatial attributes. The study constructed an OLS regression model of base station population in non-work time and nine types of building data, and the equation is as follows:

$$y = \beta_0 + \beta_1 x_1 + \beta x_2 + \ldots + \beta_n x_n + \varepsilon \, (n \le 9) \tag{1}$$

where $y$ is the mobile population, $x_n$ is the area of nine types of buildings, $\beta_0$ is the intercept, $\beta_n$ is the regression coefficient of each factor, and $\varepsilon$ is the random error.

In the BUR model, we used the scatter plot to detect the correlation between the non-work time population and nine types of buildings, the VIF parameter (with the margin of 7.5) to eliminate the collinearity of the explanatory variables, and the $P$-value to obtain the final explanatory variables that pass the 1% significance level test. The study randomly selected 80% of the data as the training set and the other 20% as the test set and used the R-square to test the fitting degree of the model.

The base station population and the building data vary with different spatial locations, demonstrating the existence of spatial non-stationarity. The global OLS model can only reflect the average level of the study area, while geographically weighted regression [38] can solve the problem of spatial heterogeneity. Consequently, the study further constructed the geographically weighted regression model (Equation (2)) of the base station population and five types of building data that passed the OLS regression test to explore the ability of building data to reflect spatial heterogeneity. The equation is as follows:

$$y_i = \beta_0(u_i, v_i) + \sum_{k=1}^{p} \beta_k(u_i, v_i) x_{ik} + \varepsilon_i \, (i = 1, 2, 3, 4, \ldots, n) \tag{2}$$

where $(u_i, v_i)$ is the spatial coordinate of the ith position, and $\beta_k \, (u_i, v_i) x_{ik}$ is the kth model regression parameter of the ith position, and $\varepsilon_i$ is the random error of the ith position.

### 2.3.3. Grid Mapping and Population Check

With consideration of the existence of spatial heterogeneity, the market penetration rate of mobile phone service operators between administrative units across the study area is different. The study counted the ratio of the total non-work time mobile phone population in each complete administrative area, based on the CDR data in Wuhan, to the permanent resident data in each complete administrative area, based on the census data of 2016.

Based on the BUR and the market penetration rate of operators, the study calculated the predicted population of grids (Equations (3) and (4)), and drew the non-work time population mapping of Wuhan Urban Development Zone in 1 km resolution. The grid mapping paves the way for the exploration of the population distribution characteristics. The equations are as follows:

$$Y_a = \frac{C}{M} Y_m \tag{3}$$

$$Y_m = \beta_i + \beta_A x_A + \beta_B x_B + \beta_H x_H + \beta_R x_R + \beta_{MW} x_{MW} + \varepsilon \tag{4}$$

where $Y_a$ is the grid population, $C$ is the resident population data in the administrative district, $M$ is the mobile population data in the administrative district, $Y_m$ is the predicted grid mobile population,

$\beta_i$ is the intercept, $\beta_A$, $\beta_B$, $\beta_H$, $\beta_R$, $\beta_{MW}$ are the coefficients of five types of buildings, $x_A$, $x_B$, $x_H$, $x_R$, $x_{MW}$ are the various building area within grids, and $\varepsilon$ is the random error.

Due to the lack of fine-grained population data, nine administrative areas that are completely contained within the study area were selected for the population check. The study summarized the predicted population of grids included in each administrative district, and performed validation through the relative error (RE, Equation (5)) and the mean prediction error (MPE, Equation (6)) with the census data of 2016 as a baseline.

$$RE = \frac{A - C}{C} \times 100\% \tag{5}$$

$$MPE = \frac{\sum_{i=1}^{m}(RE)_i}{m} (m = 9) \tag{6}$$

where A is the total predicted population of the grid in the entire administrative district, C is the resident population data of the entire administrative district, i is the ith statistical unit, and m is the number of statistical units.

## 3. Results

### 3.1. Status Data Analysis

#### 3.1.1. Distribution of Mobile Phone Population

Based on the original simple Voronoi's 9722 research units, the number of Voronoi map units generated by the grid base station was 1487. The spatial distribution of the mobile population (Figure 4) and statistics (Table 3) show that the average population in each Voronoi unit is small, and the spatial transition is stable. In contrast, the population distribution area of the Grid Voronoi unit is more integrated.

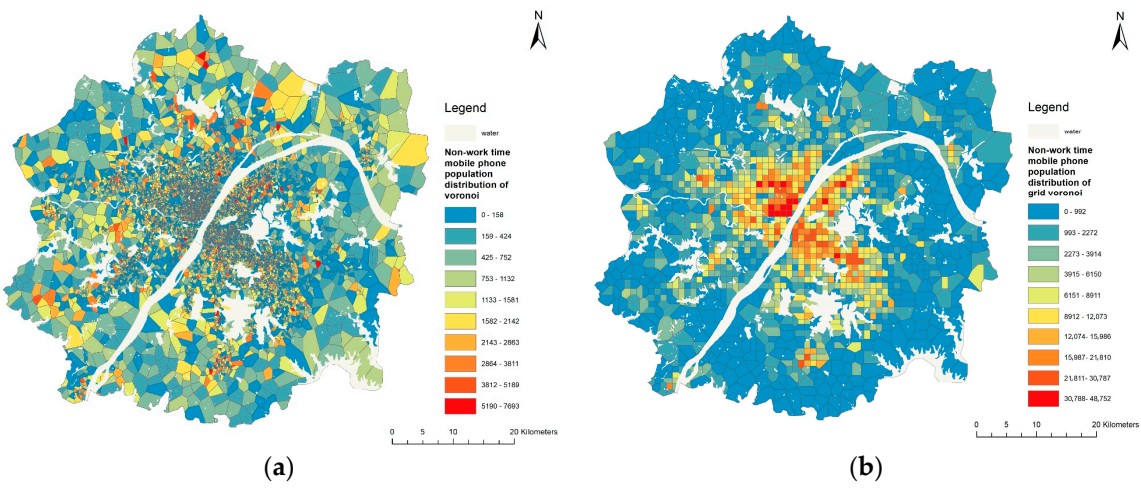

(**a**)　　　　　　　　　　　　　　　　　　(**b**)

**Figure 4.** Mobile phone population map in non-work time: (**a**) simple Voronoi; (**b**) Grid Voronoi.

**Table 3.** Demographics of mobile phones in non-work time.

|  | Min | Max | Sum | Average | Standard Deviation | Count |
|---|---|---|---|---|---|---|
| Voronoi | 0 | 7693 | 6,562,621 | 708.12 | 993.86 | 9277 |
| Grid Voronoi | 0 | 48,752 | 6,562,621 | 4413.33 | 6517.70 | 1487 |

According to the area statistics of Voronoi units (Table 4), the standard deviation of the unit area of Voronoi is lower than that of Grid Voronoi, while the range of Voronoi is far higher than that of

Grid Voronoi. The main reason is due to the extremely uneven spatial distribution of the original base stations. As the highly clustered base stations in the central urban area leads to a large number of low-value area aggregation of the original Voronoi units, the range is too large and the average area value is low, which reduces the fitting effect of building use regression. As is shown in Table 4, Grid Voronoi reduces the range of the research unit and makes its distribution more stable, which is conducive to the construction of an effective building use regression model.

**Table 4.** Unit area statistics.

|  | Min (km$^2$) | Max (km$^2$) | Sum (km$^2$) | Average (km$^2$) | Standard Deviation | Count |
|---|---|---|---|---|---|---|
| Voronoi | $1.191 \times 10^{-5}$ | 22.904 | 3261.612 | 0.352 | 1.074 | 9277 |
| Grid Voronoi | 0.949 | 23.004 | 3261.612 | 2.193 | 2.078 | 1487 |

### 3.1.2. Building Density Based on Grid Voronoi

Five types of significantly related building density statistical box plots (Figure 5) based on Grid Voronoi diagrams show that the density parameters of R buildings are higher than those of other buildings, from the average, maximum, and standard deviation of building density. It demonstrates that the spatial distribution of R-type buildings is larger and more extensive than that of other types of buildings. In contrast, the A, B, H, and MW buildings maintain a similar horizontal distribution.

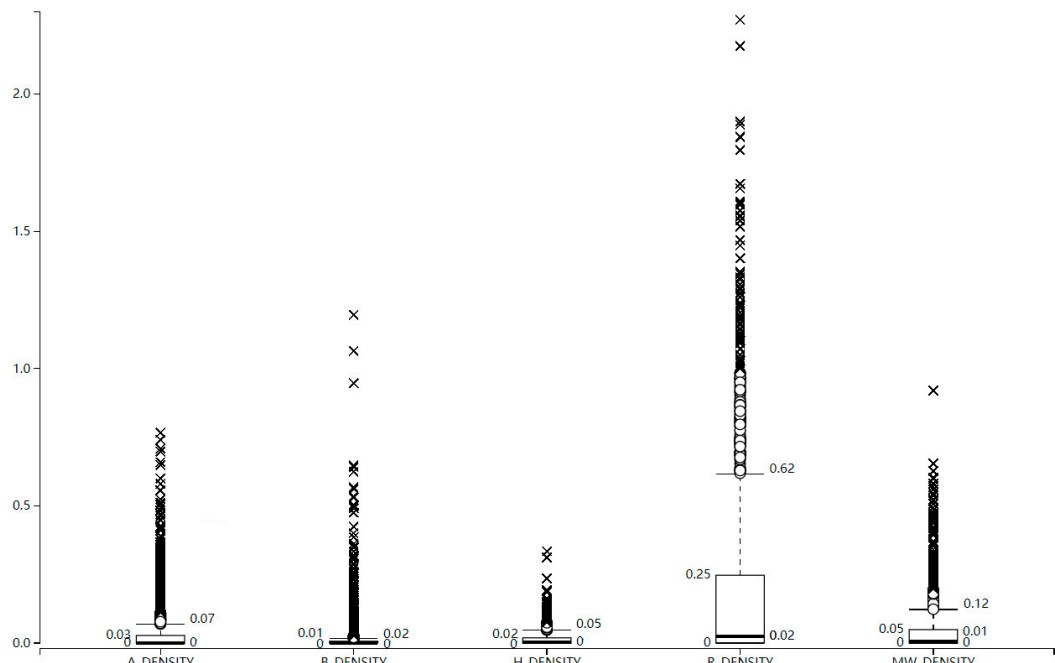

**Figure 5.** Box diagram of five types of building density.

According to the spatial visualization of five types of building density (Figure 6), the spatial distribution of class A, B, and R buildings (Figure 6a–c) is centered on the Yangtze River line in the main urban area. The building densities increase with decreasing distance from the central region. Class H buildings (Figure 6d) are mainly distributed in the periphery of the Urban Development Zone with slow development, and its overall density patterns show the low-level in the middle and high-level all around. Class MW buildings (Figure 6e) mainly distribute near the peripheral part of the central area, forming a multi-centered spatial structure, which is conducive to the development of the industry and logistics transportation.

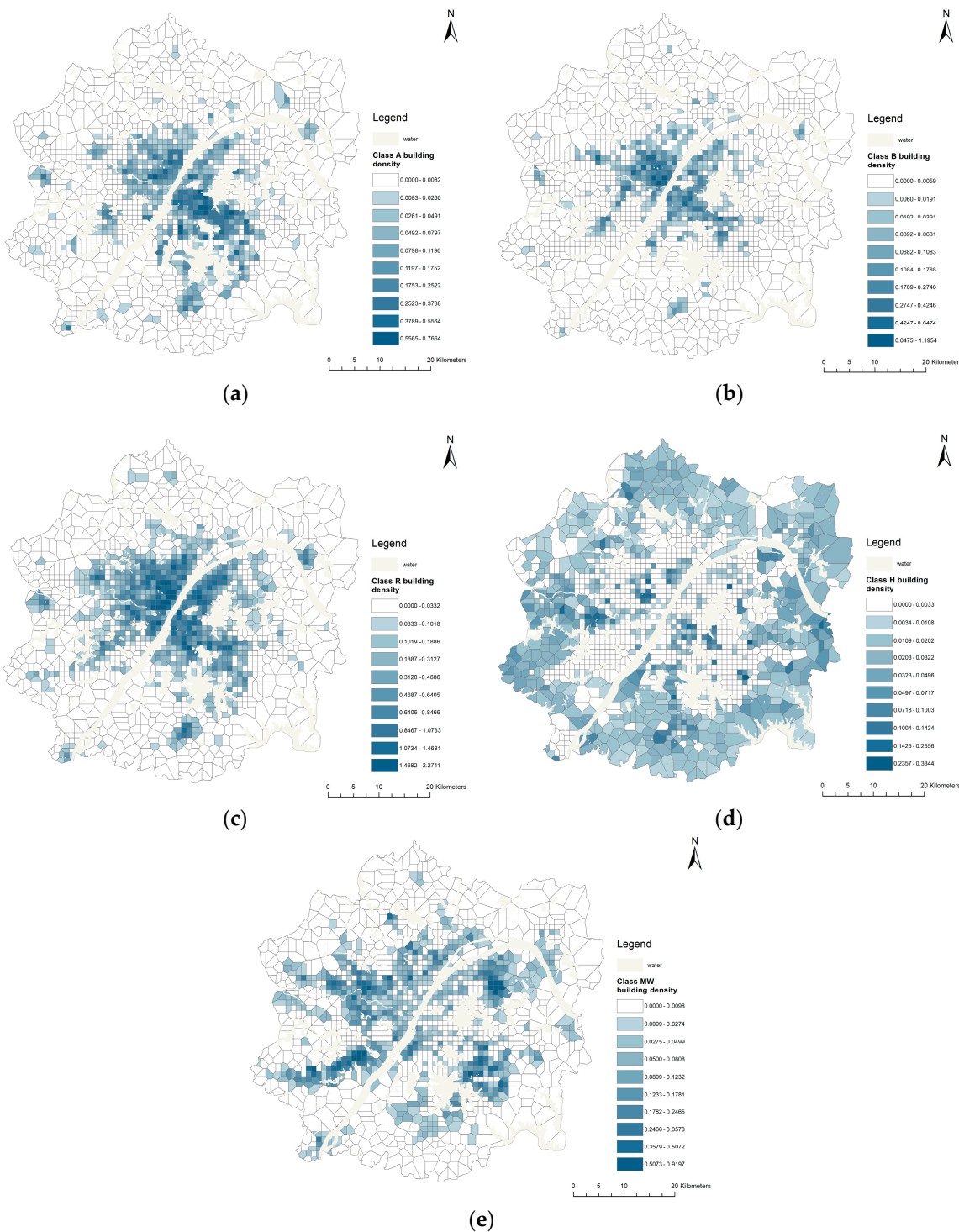

**Figure 6.** Building density statistics based on Grid Voronoi: (**a**) Class A (administrative and public service) building density; (**b**) Class B (commercial service facility) building density; (**c**) Class R (residential) building density; (**d**) Class H (development and construction) building density; (**e**) Class MW (industrial and logistics storage) building density.

*3.2. Results of Regression Models*

The scatter diagram of variables (Figure 7) shows the linear correlation between the independent variables of nine types of buildings and the population in non-work time, and it shows that class A, B, and R buildings and the population determine strong linear trends. Furthermore, five explanatory

variables, class A, B, H, R, and MW buildings, are obtained through the collinearity test and 1% significance level test (Table 5).

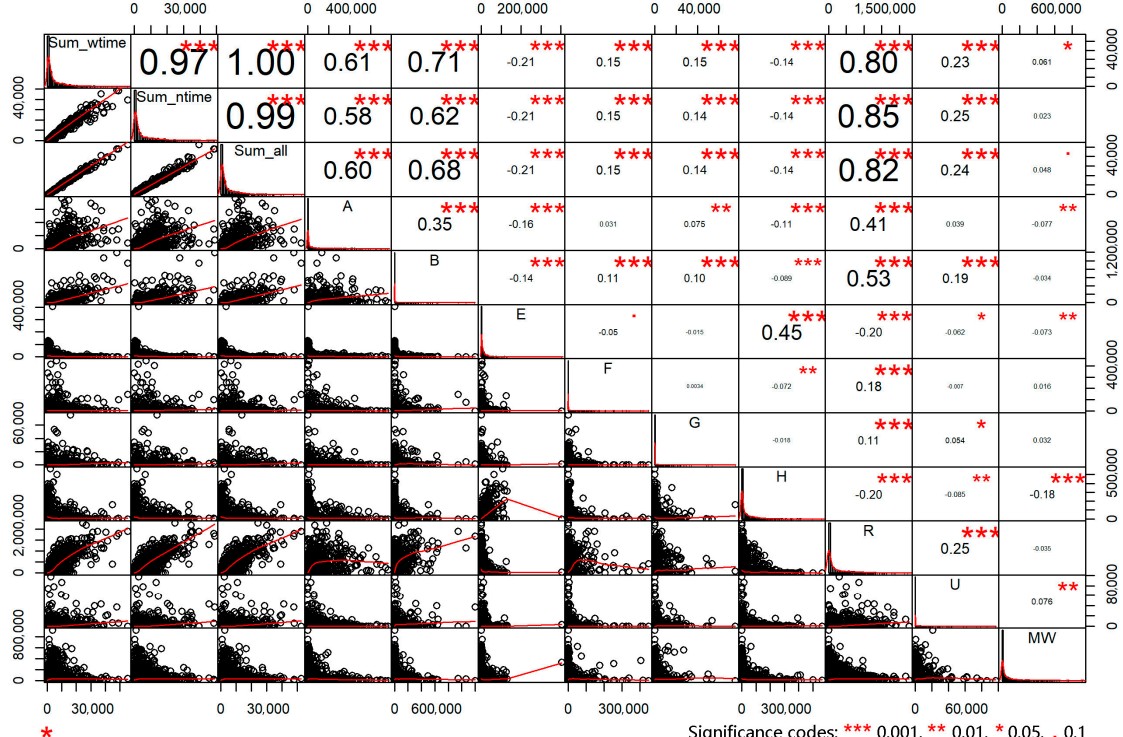

**Figure 7.** Variable scatter plot.

**Table 5.** Parameter selection.

|     | Probability [b] | Robust_Pr [b] | VIF      |
| --- | --------------- | ------------- | -------- |
| A   | 0.000000        | 0.000000      | 1.262068 |
| B   | 0.000000        | 0.000000      | 1.448988 |
| E   | 0.019296        | 0.037308      | 1.288854 |
| F   | 0.396817        | 0.494394      | 1.043214 |
| G   | 0.015922        | 0.147935      | 1.017504 |
| H   | 0.000003        | 0.000000      | 1.320904 |
| R   | 0.000000        | 0.000000      | 1.674151 |
| U   | 0.001080        | 0.065658      | 1.094622 |
| MW  | 0.000000        | 0.000000      | 1.053224 |

The adjusted R-square of the training set in the BUR regression model (Table 6) is 0.816, which indicates that the five types of buildings (A, B, R, H, and MW) are significantly correlated with the population. The R square of the data sampling validation set (Table 7) is 0.776, and the mean absolute percentage error is 2.841. In general, the Grid Voronoi building use regression method is feasible.

**Table 6.** Building use regression (BUR) diagnostics [1].

| Variable | Coefficient | Stand Error | t-Value | *p*-Value |
|----------|-------------|-------------|---------|-----------|
| Intercept | 154.826 | 114.680 | 1.350 | 0.177 |
| A | 0.018 | 0.001 | 21.225 | 0.000 |
| B | 0.014 | 0.001 | 14.111 | 0.000 |
| H | 0.005 | 0.001 | 4.037 | 0.000 |
| R | 0.012 | 0.000 | 46.796 | 0.000 |
| MW | 0.005 | 0.001 | 7.132 | 0.000 |

[1] **Number of Observations:** 1487; **AICc:** 27833.573. **Multiple R-Squared:** 0.816; **Adjusted R-Squared:** 0.816. **Note:** A (administrative and public service building); B (commercial service facility building); H (development and construction building); R (residential building); MW (industrial and logistics storage building).

**Table 7.** The validation set diagnostics.

| | |
|---|---|
| $R^2$ | 0.776 |
| **Mean Absolute Error** | 1770.032 |
| **Mean Squared Error** | 9364121 |
| **Root Mean Squared Error** | 3060.085 |
| **Mean Signed Difference** | 282.078 |
| **Mean Absolute Percentage Error** | 2.841 |

Based on the BUR, the study constructed a GWR model (Table 8) of the non-work time population and five types of building data. The corrected R square is 0.828, which indicates that the GWR model has a limited improvement on the regression level of the OLS model. In general, the building data itself can reflect spatial heterogeneity.

**Table 8.** Geographically weighted regression.

| Varname | Variable |
|---------|----------|
| Bandwidth | 9285.156 |
| Sigma | 2701.006 |
| AICc | 27753.206 |
| $R^2$ | 0.837 |
| $R^2$ Adjusted | 0.828 |

The coefficient statistics of the five types of significant explanatory variables related to the population (Figure 8) shows that the class R building coefficient fluctuates stably, since class R buildings have a more balanced effect on the population distribution space. In terms of the range statistics of coefficients, class A, H, R, and MW buildings always show positive spatial correlations with the population. Class B buildings show a trend of alternating positive and negative correlations, with a lower frequency of negative values. The medians of class A, B, and R buildings are higher, and those of class H and MW buildings are lower. In general, coefficient values reflect the ability of the five types of buildings to affect the population distribution.

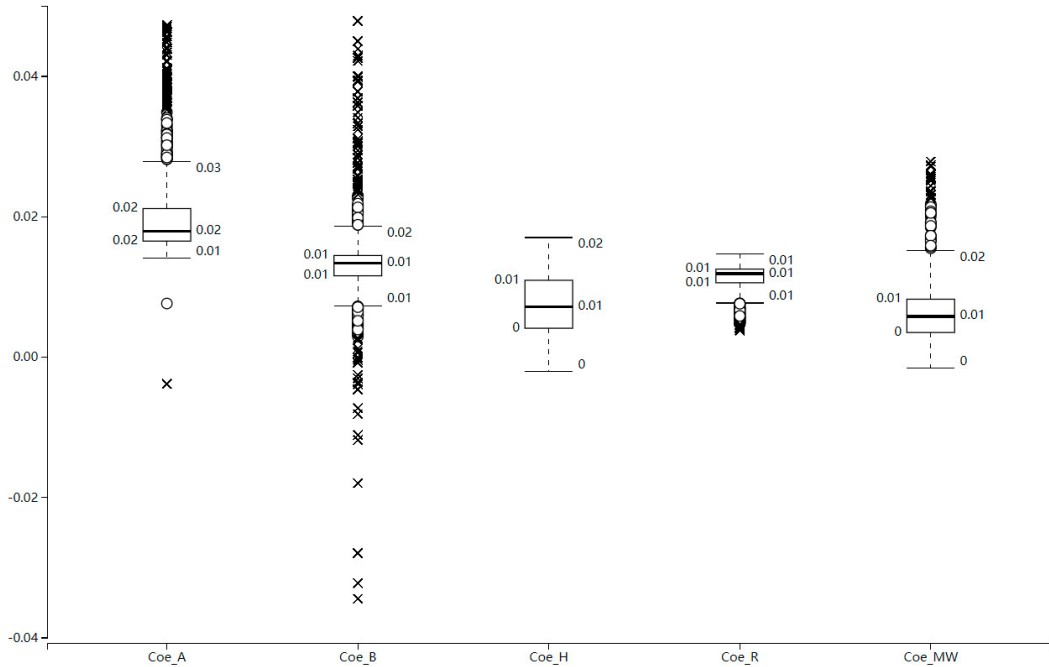

**Figure 8.** Box diagram of variable coefficients.

Coefficient spatial visualization further explains the influence mechanism of building properties on population mobility and spatial distribution. Class A buildings (Figure 9a) have a significant positive correlation impact on the population in Jiangxia District in the south, and Xinzhou District in the northeast of the study area. In these areas, the public service and commercial facilities are relatively complete, which can promote the population distribution to a certain extent.

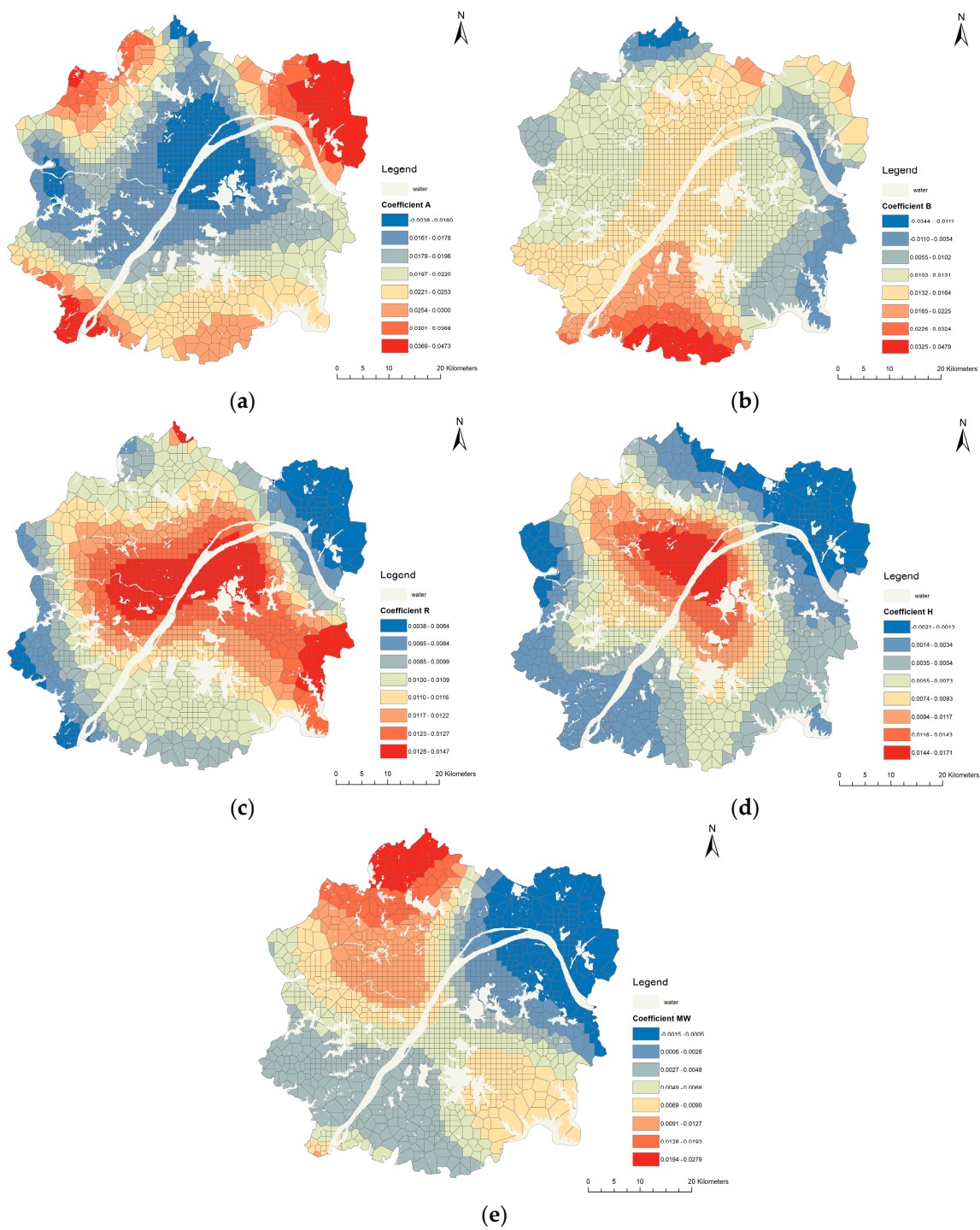

**Figure 9.** Spatial statistics of variable coefficients: (**a**) Coefficient A (administrative and public service building); (**b**) Coefficient B (commercial service facility building); (**c**) Coefficient R (residential building); (**d**) Coefficient H (development and construction building); (**e**) Coefficient MW (industrial and logistics storage building).

Jiangxia District has vigorously developed and improved commercial facilities in recent years, hence class B buildings (Figure 9b) have a more apparent positive effect on population distribution in this area.

The distribution of the correlation coefficient of the class R buildings (Figure 9c) generally shows a high level in the middle of the main city center and gradually decreases outwards, which indicates

that the positive contribution value of residential buildings in the central city area to the population distribution is more evident than that in other regions.

Since the growth rate of construction in Jianghan District and the north of Hongshan District is increasing, the correlation coefficient of class H buildings (Figure 9d) decreases outwards centered on these regions. As a whole, the class H buildings show a positive impact on population distribution.

Huangpi District in the northwest is mainly engaged in industrial development, namely logistics and warehousing, and is also the location of Tianhe Airport. Therefore, the positive correlation between class MW buildings (Figure 9e) and population in the region is more significant than that in other regions.

### 3.3. Grid Mapping and Population Check

The administrative district population check based on the operator's market penetration rate (Table 9) shows that the MPE of the predicted population data is 9.72%. Among the nine administrative region samples used for verification, the absolute value of RE of the predicted population is less than 30%, and the numbers of administrative regions below 5%, 15%, and more than 15% are 3, 8, and 1, respectively. The absolute value of RE of the predicted population in Caidian District is the highest, with the relatively higher predicted population value. The reason may be that the regional residential development has been fast in recent years, while the occupancy rate is low. Consequently, the biased population distribution appears in the region. In general, the proposed building use regression model can well improve the spatial granularity of population distribution, with high accuracy and feasibility.

**Table 9.** Grid population check based on the operator's market proportion.

| Name | Resident Population | Mobile Population | Predicted Non-Work Time Population | RE (%) | MPE (%) |
|---|---|---|---|---|---|
| Caidian | 719,900 | 393,577 | 934,861 | 29.86 | |
| Dongxihu | 541,100 | 429,287 | 516,816 | −4.49 | |
| Hanyang | 648,500 | 488,036 | 705,456 | 8.78 | |
| Hongshan | 1,609,900 | 1,276,304 | 1,760,632 | 9.36 | |
| Jiang'an | 961,300 | 664,761 | 994,017 | 3.40 | 9.72 |
| Jianghan | 729,500 | 528,006 | 686,765 | −5.86 | |
| Qiaokou | 867,100 | 488,553 | 773,192 | −10.83 | |
| Qingshan | 526,800 | 277,512 | 589,607 | 11.92 | |
| Wuchang | 1,274,000 | 864,964 | 1,313,014 | 3.06 | |
| Hannan | 131,600 | 66,682 | 86,470 | | |
| Jiangxia | 894,600 | 780,585 | 916,490 | | |
| Huangpi | 967,100 | 515,291 | 391,201 | | |
| Xinzhou | 894,800 | 342,955 | 461,990 | | |

The further grid population mapping (Figure 10) shows the highly clustering pattern, mainly centered on Wuchang, Hankou, and Hanyang, the central urban regions within the Third Ring Road, and with the Yangtze River as the center for linear expansion. In this area, the water body plays a crucial role in separating the population distribution in the city, and the location of the new Yangtze River City, with Chenjiaji as the starting area, is more conducive to optimizing the Yangtze River axis and promoting the overall balanced development of Wuhan.

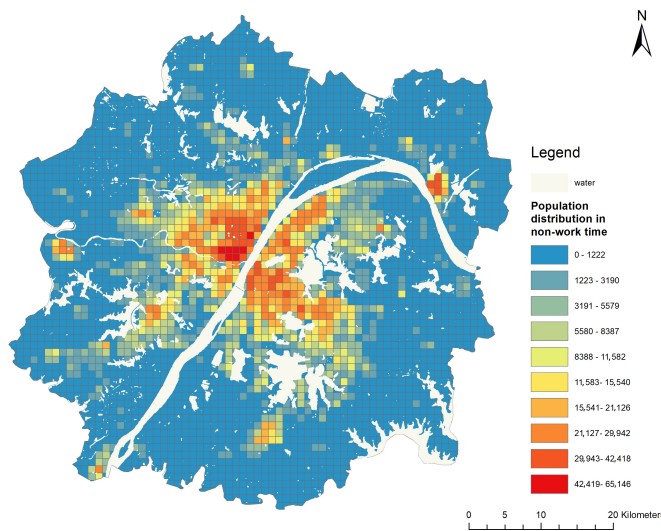

**Figure 10.** Population distribution in non-work time.

On the right bank of the Yangtze River, outside the Third Ring Road, the Zhifang area in Jiangxia District, the northern Guanggu area in Hongshan District and the Thomson Lake area, have gradually achieved the coordinated development, with densely distributed population. In contrast, the southern end of Jiangxia District has a small population, due to its slow economic development. Qingshan District, which is the seat of Wuhan Iron and Steel Group, and Xinzhou District in the northeast, with comprehensive public service facilities such as colleges and universities, have relatively large populations.

On the left bank of the Yangtze River, outside the Third Ring Road, the southern Hougong Lake area and the Triangular Lake area are gradually developing close connections with the Hanyang District in the north, with a dense population. In the west, people mainly concentrate in the Caidian District, with relatively complete public service facilities. However, it is less connected to the main urban area, and has not achieved continuous development. In the north, the population density around the Jinyin Lake area in the Dongxihu District and the southern region in Huangpi District are high, due to the dense distribution of universities, airports, and other facilities.

## 4. Discussion

The application of spatiotemporal sources like mobile phone data for mapping human population distribution in the urban area is mushrooming. Mobile phone data has a significantly high penetration rate across the urban area, which represents relatively reliable information on real-time population patterns. The CDR data paves the way for achievements in fine-scale dasymetric population mapping. A variety of approaches have been published. For instance, Deville et al. [22] introduced explicit estimations of national-scale population densities with the usage of mobile phone data in Portugal and France. Kubíček et al. [39] proposed a building level dasymetric approach to evaluate the spatiotemporal distribution of population derived from mobile phone data in Brno, Czech Republic. However, there are some challenges that are usually less acknowledged or ignored. This study pays attention to one of the challenges: the extremely uneven configuration of base stations with their high-uncertain serving scopes. Based on Voronoi, the theoretical coverage scopes of base stations, the study proposes source and target zones of the spatial disaggregation of mobile phone data, similar to [40], in which a multi-temporal, function-based dasymetric interpolation method was proposed. Specifically, we differ in the following aspects: (1) We generate the Grid Voronoi diagrams by aggregating base stations into 1 km grids. Compared with simple Voronoi diagrams, Grid Voronoi can improve the overall spatial distribution of base stations and reduce the negative impact of excessively large simple Voronoi unit area differences; (2) We further develop the building data at the

administrative level. Individual buildings are finely-divided into nine more detailed types by land use data, and the corresponding areas are obtained. (3) The OLS regression model and the GWR model are constructed from the spatial homogeneity and spatial heterogeneity, to discuss the ability of the building data to reflect spatial heterogeneity.

The empirical findings show that the applied Grid Voronoi building regression model, with careful consideration of the built environment attributes, is of high accuracy and feasibility. The model can refine the population from the CDR data to desired target zones (1 km grids). The R-square values of the training set and verification set are 0.816 and 0.776, respectively. Moreover, the selected five types of buildings are A (administrative and public service building), B (commercial service facility building), H (development and construction building), R (residential building), and MW (industrial and logistics storage building), respectively. The coefficient R is far above others, since the main activities and mobilities of human beings are around residential buildings during the non-work time. The population evaluation of the BUR method is further verified by the population check, based on the market penetration of operator services. The absolute value of RE is generally less than 30%, and the MPE is 9.72%, demonstrating the applicability of the BUR model to obtain population distribution information in Wuhan.

The study did not further discuss how to apply the Grid Voronoi building use regression model for social issues. The more timely and reliable population distribution map over time can be provided through the BUR model by decomposing mobile phone data into detailed grid population data. Consequently, the integration of climate data [41], crime data [42], location data [39] of drinking water tanks, and other urban social spatial data with grid population can offer the possibility to perform object-oriented risk assessments.

In general, this study remains challenging and limited. Firstly, although the study uses the detailed continuous-time mobile phone population data, including the total population in work time, non-work time, and all-time, it is only within one month. The long-term detailed call data will be able to provide a more accurate reference for the spatiotemporal population distribution in the city. Then, in terms of scope on a larger scale, it remains difficult for data acquisition and quantity, as well as requirements for computers with high software calculation capabilities.

Furthermore, the Grid Voronoi building use regression model proposed by the research can be expanded in the following aspects in the future: (1) further constructions of work time and all-time building regression models with the population derived from mobile phone data to deeply understand the spatiotemporal population dynamic distribution in urban space; (2) with the constantly increasing availability of semantic 3D city models [43], the vertical attribute can be included into the attributes of the building data to study the correlation between different building space volumes and population distribution from a three-dimensional perspective; and (3) empirical researches on the potential combinations of different scale grids with building data and mobile phone data in BUR models, as well as their impact on results.

## 5. Conclusions

High-precision population mapping is a crucial component of the fine model of urban development. This study contributes a proposition for the application of a Grid Voronoi building use regression model to attain the gridded human spatial population in the development area of Wuhan. We redefine the service scope area of base stations by aggregating stations into 1 km grids to reduce the negative impact of excessively large simple Voronoi unit area differences. In the BUR, population derived from mobile phone data, which can reflect relatively reliable information on real-time population distribution, and land use integrated building data are used for the training set, with census data for verification. As the empirical finding shows, the GWR model has a limited level of improvement on the OLS regression model, demonstrating that the building data itself can well reflect spatial heterogeneity.

In general, the building use regression model can reveal the close correlation between population distribution, building area, and usage characteristics, and provide the fresh and ideal perspective for drawing high granularity population maps. The fine-scale dasymetric population mapping is of great significance

for the effective identification of urban population spatial distribution characteristics, the discovery of urban problems, and the improvement of urban management, as well as the formulation of urban policies.

**Author Contributions:** Data curation: Zhenghong Peng and Hao Wu; formal analysis: Zhenghong Peng; investigation: Hao Wu; methodology: Ru Wang and Lingbo Liu; writing—original draft: Ru Wang; writing—review and editing: Lingbo Liu. All authors have read and agreed to the published version of the manuscript.

**Funding:** The study was funded by National Natural Science Foundation of China (No. 51978535); Humanities and Social Science Project of the Ministry of Education (No. 19YJCZH187); Wuhan University Experiment Technology Project Funding.

**Acknowledgments:** The authors acknowledge the contribution of all the anonymous reviewers that improved the quality of the paper.

**Conflicts of Interest:** The authors declare no conflict of interest.

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
