# Peer review of "Fine-Scale Dasymetric Population Mapping with Mobile Phone and Building Use Data Based on Grid Voronoi Method"

_ijgi, doi:10.3390/ijgi9060344_

Round 1
Reviewer 1 Report
The research can be published after the revisions authors have made.
Author Response
Dear reviewer :
Really thanks for your helpful comments. We have checked the manuscript and revised it according to the comments.
Sincerely yours,
Ru Wang
Reviewer 2 Report
I would like to thank the authors of this article that I found interesting and full of innovative ideas. Certainly the analysis depends heavily on the availability of mobile phone data that is not the same in all parts of the world.
In general, I consider the article suitable for publication following some minor revisions.
The authors already announce in the abstract the advantages of using mobile phone data compared to more traditional census data. They propose an integrated method between voronoi grid and disimetric maps to define the spatial boundaries of population data acquired through mobile phones.
In the first paragraph the topic of disimetric maps is introduced in a clear and synthetic way so that even the less expert in the field can understand what we are talking about. The negative (resolution dependence) and positive aspects are highlighted and the solution or the overcoming of these limits with the grid voronoi and building use regression method is proposed.
In general, please pay attention to the use of very long sentences that confuse the reader.Please check the acronyms, even if they are widely used terms you should make them explicit before entering the acronym.For example Ordinary Least Squares (OLS) is not explicit and in L 178 and 179 first the extended form and then the acronym.
Structure:
There are too many sub paragraphs I suggest we rework the structure of the paper. Here are some suggestions:
- merge paragraphs 3.1.1 and 3.1.2 into one paragraph 3.1 Data and preprocessing. The results paragraph also contains many sub-paragraphs that are generally confusing to read.
- Study area: very very short paragraph, consider including it in the data description paragraph.
Other comments:
- L 93-98 very very long sentence, please rephrase. I would also recommend that you enter the classification of the data in the form of a table to make it more readable.
- I do not understand Figure 2 a and 2 b. If as announced in L 100-105 they contain various classes of building use, a legend should be inserted in the figure to make the maps more understandable.
- L 104 In the sentence "...E (non-construction land, including waters, agricultural and forest land and other non-construction land)..." what is the difference between "non-construction land" and "other non-construction land"?
- L 109-110 please rephrase, it is not clear.
- In the collinearity test what margin do you use for the VIF and what is the result of the collinearity test (i.e. the VIF value)? this should be specified in the methodology and then in the results
- L 343-344 "Literature..." I think it's better to write the name of the author for example: "...Devid, P. et al [22]"
Author Response
Dear reviewer 2:
Really thanks for your helpful and detailed comments. We have responded to each of your comments,please see the attachment.
Sincerely yours,
Ru Wang

Reviewer 3 Report
This is an interesting application on fine-scale population mapping. However, some issues need to be addressed:
a) It is not clear in the study of how validation of the proposed methodology was conducted. What was the date of the census data? How many model runs were done during the regression?
b) Information concerning the number of base-stations used, and the number of base stations per grid is not clearly provided. Furthermore, information regarding the link between a base station and the number of telephone users it serves is not clearly explained.
c) The figures/maps are illegible while others lack informative legends and need to be improved
Thank you
Author Response
Dear reviewer :
Really thanks for your helpful and detailed comments. We have responded to each of your comments. Please see the attachment.
Sincerely yours,
Ru Wang

Round 2
Reviewer 3 Report
I would like to thank the authors for this improved version and the detailed response to the queries that were raised initially.
Thank you.